# Urinary cell-free DNA is a versatile analyte for monitoring infections of the urinary tract

Philip Burnham[1], Darshana Dadhania[2,3], Michael Heyang[1], Fanny Chen[1], Lars F. Westblade[4,5], Manikkam Suthanthiran[2,3], John Richard Lee[2,3] & Iwijn De Vlaminck [1]

Urinary tract infections are one of the most common infections in humans. Here we tested the utility of urinary cell-free DNA (cfDNA) to comprehensively monitor host and pathogen dynamics in bacterial and viral urinary tract infections. We isolated cfDNA from 141 urine samples from a cohort of 82 kidney transplant recipients and performed next-generation sequencing. We found that urinary cfDNA is highly informative about bacterial and viral composition of the microbiome, antimicrobial susceptibility, bacterial growth dynamics, kidney allograft injury, and host response to infection. These different layers of information are accessible from a single assay and individually agree with corresponding clinical tests based on quantitative PCR, conventional bacterial culture, and urinalysis. In addition, cfDNA reveals the frequent occurrence of pathologies that remain undiagnosed with conventional diagnostic protocols. Our work identifies urinary cfDNA as a highly versatile analyte to monitor infections of the urinary tract.

[1] Meinig School of Biomedical Engineering, Cornell University, Ithaca, NY 14853, USA. [2] Division of Nephrology and Hypertension, Department of Medicine, Weill Cornell Medicine, New York, NY 10065, USA. [3] Department of Transplantation Medicine, New York Presbyterian Hospital–Weill Cornell Medical Center, New York, NY 10065, USA. [4] Department of Pathology and Laboratory Medicine, Weill Cornell Medicine, New York, NY 10065, USA. [5] Division of Infectious Diseases, Department of Medicine, Weill Cornell Medicine, New York, NY 10065, USA. These authors contributed equally: John Richard Lee, Iwijn De Vlaminck. Correspondence and requests for materials should be addressed to J.R.L. (email: jrl2002@med.cornell.edu) or to I.D.V. (email: vlaminck@cornell.edu)

Urinary tract infection (UTI) is one of the most common medical problems in the general population[1]. Among kidney transplant recipients, UTIs occur at an alarmingly high rate[2]. Bacterial UTI affects approximately 20% of kidney transplant recipients in the first year after transplantation[3] and at least 50% in the first 3 years after transplantation[4]. In addition, complications due to viral infection often occur. An estimated 5–8% of kidney transplant recipients suffer nephropathy from BK polyomavirus (BKV) infection in the first 3 years after transplantation[5,6]. Other viruses that commonly cause complications in kidney transplantation include adenovirus, JC polyomavirus, cytomegalovirus (CMV), and parvovirus[7]. The current gold standard for diagnosis of bacterial UTI is in vitro urine culture[8]. Although improved culture methods are being investigated[9,10], bacterial culture protocols implemented in clinical practice remain limited to the detection of relatively few cultivable organisms[9]. Furthermore, urinalysis is often required in conjunction with culture to make treatment decisions[8]. A large number of small fragments of cell-free DNA (cfDNA) are present in plasma and urine[11–14]. These molecules are the debris of the genomes of dead cells and offer opportunities for precision diagnostics based on 'omics principles, with applications in pregnancy, cancer, and solid-organ transplantation[13,15–17].

Here we investigate the utility of urinary cfDNA to comprehensively monitor host and pathogen interactions that arise in the setting of viral and bacterial infections of the urinary tract. Using shotgun DNA sequencing, we assay cfDNA isolated from 141 urine samples collected from a cohort of 82 kidney transplant recipients, including recipients diagnosed with bacterial UTI and BKV nephropathy (BKVN). We developed a single-stranded DNA (ssDNA) library preparation, optimized for the analysis of short, highly fragmented DNA[18–20], and were able to sequence cfDNA isolated from relatively small volumes of urine supernatant (1 mL or less). We find that urinary cfDNA sequencing agrees in the vast majority of cases with conventional diagnostic testing, while also uncovering frequent occurrence of bacteria and viruses that remain undetected in conventional diagnostic protocols. We further investigated cfDNA-based analytic methods that go beyond microbial identification and provide a deeper understanding of the infectious process. We show that rate of bacterial population growth can be estimated from an analysis of the bacterial genome structure and that this measurement can inform diagnosis of UTI. We furthermore mined cfDNA for antimicrobial resistance (AR) genes and show that AR gene profiling can be used to evaluate AR. Last, we observe that the relative proportion of kidney donor-specific cfDNA correlates with graft tissue injury in the setting of viral infection and host immune cell activation in the setting of bacterial infection.

Collectively, our study supports the use of shotgun DNA sequencing of urinary cfDNA as a comprehensive tool for monitoring patient health and studying host–pathogen interactions.

## Results

**Biophysical properties of urinary cfDNA.** Urinary cfDNA is composed of human chromosomal, mitochondrial, and microbial cfDNA released from host cells and microbes in the urinary tract and of plasma-derived cfDNA that passes from blood into urine[21]. Urine can be collected non-invasively in large volumes and therefore represents an attractive target for diagnostic assays. Compared to plasma cfDNA, relatively few studies have examined the properties and diagnostic potential of urinary cfDNA. The urinary environment degrades nucleic acids more rapidly than plasma resulting in fewer DNA fragments that are also

shorter[22]. Consequently, sequence analyses of urinary cfDNA have to date required relatively large (>10 mL) volumes of urine[14,23]. Here we applied a single-stranded library preparation technique that employs ssDNA adapters and bead ligation to create diverse sequencing libraries that capture short, highly degraded cfDNA[19,20] (Fig. 1a). We find that single-stranded library preparation enables sequence analyses of urinary cfDNA from just 1 mL of urine supernatant. We tested 141 urine samples collected from 82 kidney transplant recipients, including subjects diagnosed with bacterial UTI and BKVN (overview of post-transplant dates and categories depicted in Fig. 1b, see Methods). We obtained 43.5 +/− 17.3 million paired-end reads per sample, yielding a per-base human genome coverage of 0.49× +/− 0.24×. Many fragments were derived from microbiota; for example, for subjects diagnosed with bacterial UTI, bacterial cfDNA accounted for up to 34.7% of the raw sequencing reads, and in cases of BKVN, BKV cfDNA accounted for up to 10.3% of raw sequencing reads. To account for technical variability and sources of environmental contamination during extraction and library preparation, a known-template control sample was included in every sample batch and sequenced (see Methods).

We analyzed the fragment length profiles of urinary cfDNA at single-nucleotide resolution using paired-end read mapping[11]. This analysis confirmed previous observations of the highly fragmented nature of urinary cfDNA compared to plasma cfDNA[23] (Fig. 1c). We observed a 10.4 bp periodicity in the fragment length profile of chromosomal cfDNA (Fourier analysis, Fig. 1c, inset), consistent with the periodicity of DNA–histone contacts in nucleosomes[24]. BKV is known to hijack histones of infected host cells and to form mini-chromosomes after infection[25]. The periodicity in the fragment length profiles of BKV cfDNA in urine reflect this biology (Fig. 1c). We did not observe a similar nucleosomal footprint for bacterial and mitochondrial cfDNA or cfDNA arising from parvovirus B19, which is expected given the non-nucleosomal compaction of the genomes that contribute these cfDNA types (Fig. 1d).

**Infectome screening.** We assessed the presence of cfDNA from bacterial and viral pathogens reported by conventional diagnostic assays. We used bioinformatics approaches[26] to estimate the relative genomic representation of different species (see Methods). To directly compare the microbial abundance across samples, we computed the representation of microbial genome copies relative to human genomes copies and expressed this quantity as relative genome equivalents (RGE).

We detected a very high load of BKV cfDNA in all 25 samples collected from 23 subjects diagnosed with BKVN by needle biopsy (mean $1.49 +/− 1.08 \times 10^5$ RGE, Fig. 2a) but not in 10 samples from 10 subjects that were BKVN negative per biopsy (all below detection limit). In these 35 biopsy-associated samples, the BKV cfDNA abundance (RGE) correlated with a matched urine cell pellet BKV VP1 mRNA copy measurement that we previously validated as a noninvasive marker for BKVN[27,28] (Spearman's rho = 0.74, $p = 3.48 \times 10^{-7}$).

We quantified bacterial urinary cfDNA in 43 urine samples from 31 subjects who had a corresponding positive urine culture obtained on the same day (see Methods for definition). For 41 of the 43 positive urine specimens, a bacterial organism was reported to the species level by conventional culture. In 40 of these 41 samples, sequencing of urinary cfDNA detected the clinically reported organisms to the species level (Fig. 2b). For a single sample, urinary cfDNA did not match with the bacterial culture: *Raoultella ornithinolytica* was isolated in culture but not detected by cfDNA sequencing (see Methods for a detailed discussion of this discordant readout). For 2 of the 43 clinically

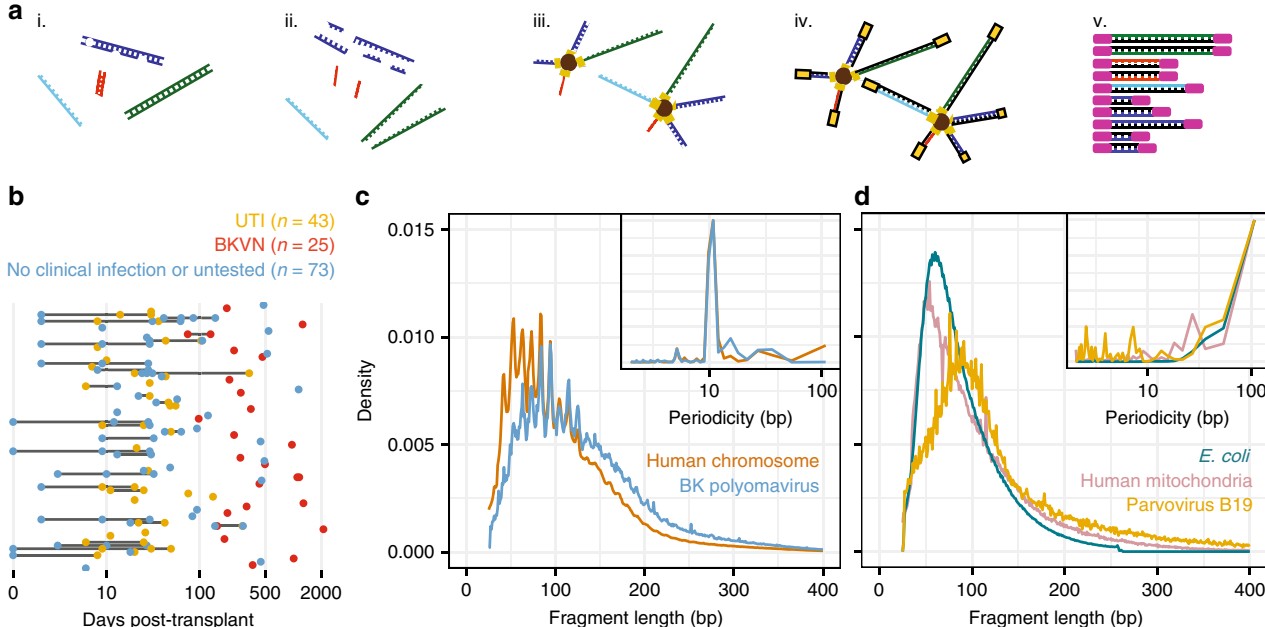

**Fig. 1** Shotgun sequencing assay and biophysical properties of urinary cfDNA. **a** Schematic representation of the ssDNA library preparation protocol used for shotgun sequencing of urinary cfDNA[20]. Key steps include: (i) cfDNA isolation, (ii) DNA denaturation, (iii) ssDNA adapter ligation, (iv) extension and double-stranded DNA adapter ligation, and (v) PCR. **b** Overview of post-transplant sample collection dates (color indicates pathology, bars connect samples from same subjects). **c**, **d** Fragment length density plot measured by paired-end sequencing for different cfDNA types: **c** chromosomal and BKV cfDNA from representative samples, and **d** *E. coli*, parvovirus B19, and mitochondrial cfDNA from representative samples. Fourier analysis reveals a 10.4 bp periodicity in the fragment length profiles of chromosomal and BKV cfDNA but not for *E. coli*, parvovirus B19, or mitochondrial cfDNA (insets). See Supplementary Data 1

positive samples, the suspected etiologic agent was identified to the genus level (*Staphylococcus*, reported as coagulase-negative *Staphylococcus* species [CoNS], and *Streptococcus*, reported as viridans group streptococci) by culture. In both these cases, the reported organism was detected as the most prevalent within the sample. From the 43 cultures, we examined 6 with polymicrobial bacterial infection (defined as 2 individual bacterial taxon detected at the genus or species level). For five out of six of these cases, we observed both species among the ten most abundant species by cfDNA sequencing. In one sample, the secondary bacterial agent, CoNS (<10,000 colony-forming units [cfu]/mL), was not detected.

To further assess the performance of urinary cfDNA for microbial identification, we compared the relative genomic abundance (RGE) of bacterial cfDNA for subjects diagnosed with bacterial infection (49 bacterial isolates identified from 43 clean-catch midstream cultures) to the relative genomic abundance (RGE) measured for 43 negative clean-catch midstream urine cultures (Fig. 2c and Supplementary Fig. 1). We found very good agreement between urinary cfDNA and culture-based isolation of *Enterococcus faecalis* (number of matched positive cultures, $n = 11$, area under the curve [AUC] = 0.97, 95% confidence interval [CI] = 0.935–1), *Enterococcus faecium* ($n = 2$, AUC = 0.98, CI = 0.976–1), *Escherichia coli* ($n = 21$, AUC = 0.97, CI = 0.93–1), *Klebsiella pneumoniae* ($n = 3$, AUC = 1.00, CI = 1), *Pseudomonas aeruginosa* ($n = 3$, AUC = 1.00, CI = 1), CoNS ($n = 4$, AUC = 0.78, CI = 0.46–1). Here receiver operator characteristic (ROC) analysis was performed for bacterial species where there was at least one positive culture of the same organism available ($n > 1$).

In only 60% of the examined samples (26/43 UTI cases) was the organism identified in culture the most prevalent organism in the sample as measured by cfDNA (Fig. 2b). Whereas bacterial

culture is skewed toward species that are readily isolated on routine bacteriological media employed for urine culture, cfDNA sequence analyses permit the identification of a broader spectrum of bacterial species. To evaluate this concept further, we assayed two samples collected from one of the subjects included in the analysis above diagnosed with *Haemophilus influenzae* bacteruria. *H. influenzae* is a very uncommon uropathogen that does not routinely grow on media employed for conventional urine culture (tryptic soy agar with sheep blood and MacConkey agar)[29]. Repeated urine cultures for this patient were negative, but given a urinalysis suggestive of a UTI and given that the patient developed *H. influenzae* bacteremia, the original urine specimen collected at presentation was re-plated onto chocolate agar, upon which *H. influenzae* was isolated. In the sample taken at the time of presentation, which was cultured, and also a sample taken 4 days after presentation, we observed *H. influenzae* cfDNA (0.037 RGE and 0.41 RGE, respectively). This case supports the utility of urinary cfDNA to identify infections where conventional culture fails.

**Profiling the urinary microbiome**. The urinary tract was regarded as sterile but recent studies have revealed that it often harbors microbiota[10,30,31]. We examined the composition of the urinary microbiome by urinary cfDNA profiling (absence of UTI, $n = 43$) or collected within the first 3 days post-transplant ($n = 12$) (Supplementary Fig. 2). We found that the species-level abundance and the species-level diversity of the bacteriome are a function of the transplant recipient gender but not the donor gender. On average, we observed two to three orders of magnitude more cfDNA from *Gardnerella* (6125×), *Ureaplasma* (1686×), and *Lactobacillus* (321×) across female transplant recipients who did not have a UTI at the time of sampling compared

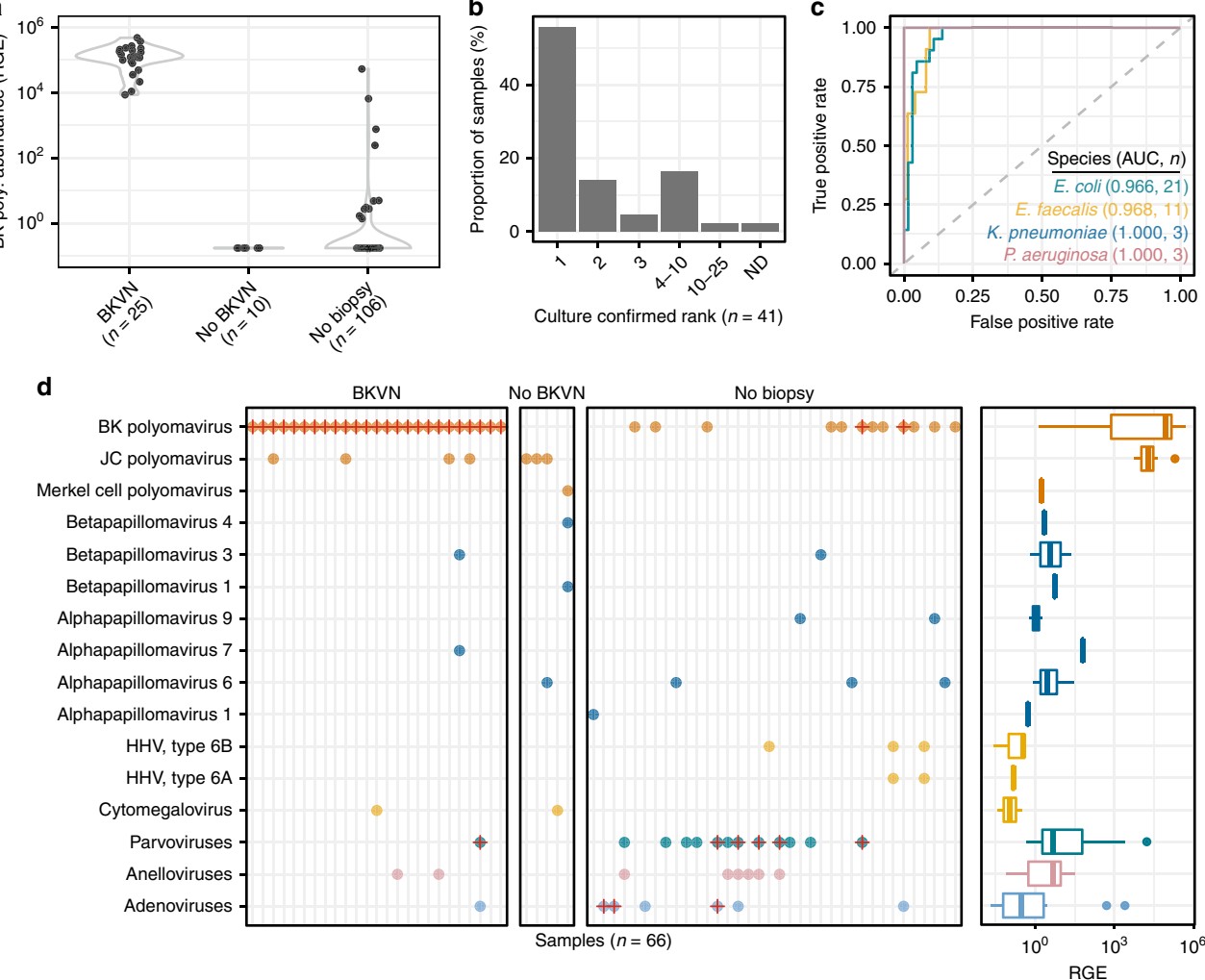

**Fig. 2** Urinary cfDNA infectome screening. **a** Violin plots of BKV cfDNA sequence abundance (in RGE) for subjects with and without BKVN and untested subjects. **b** cfDNA rank order abundance for clinically reported organisms. In 60% of samples, the bacterial organism detected in culture was the most abundant component of the cfDNA urinary microbiome (rank 1). In one sample, the clinically reported agent was not detected (ND, *R. ornithinolytica*). **c** ROC analysis of the performance of urinary cfDNA in identifying bacterial organisms (86 urine samples, AUC area under the curve, *n* number of positive cultures, see Supplementary Figure 1 for individual ROC curves for these and two additional bacteria). **d** Viral cfDNA was detected in 66 samples of the 141 samples. cfDNA reveals frequent occurrence of viruses that are potentially clinically relevant (left panel); crosses identify samples belonging to subjects who developed an infection of the corresponding viral agent. Right panel shows boxplots of the viral cfDNA abundance across all samples (right panel). Coloring of points and boxplots by viral taxonomic group; see Supplementary Data 2–5

to male recipients who did not have UTI; these bacterial genera have been characterized as microbial components of the vaginal microbiome[32]. We examined the relationship between urine collection methods and the abundance and diversity of the bacteriome and found a notably reduced bacterial load for samples collected by indwelling catheter (samples collected within 4 days after transplant) versus clean-catch urine samples. cfDNA may be an ideal tool to study the urinary microbiome, but future studies need to account for the effects of gender and sample collection approaches.

**Broad screening for viruses via cfDNA**. We next screened for the occurrence of cfDNA derived from viruses. Nearly half of the samples (66/141) had detectable levels of cfDNA derived from eukaryotic viruses that are potentially clinically relevant. Figure 2d highlights the frequent occurrence of JC polyomavirus, parvovirus B19, Merkel cell polyomavirus, CMV, human herpesvirus 6A, human herpesvirus 6B, and various known

oncoviruses across different patient groups. In several samples, we detected cfDNA from multiple polyomavirus species concurrently (JC polyomavirus or BKV). To shed light on the potential clinical utility of broad screening for viruses via cfDNA, we assayed serial urine from three subjects diagnosed with viral infections that are relatively uncommon in kidney transplant recipients and consequently not routinely screened for in our patient cohorts. In samples from two subjects with clinically diagnosed parvovirus B19 infection, we detected urinary cfDNA from parvovirus B19 8 days prior to the clinical diagnosis in one subject and urinary parvovirus B19 cfDNA 80 days before diagnosis and 25 days after diagnosis in another subject. In the former subject, we observed a high abundance of both BKV ($3.54 \times 10^4$ RGE) and parvovirus B19 ($2.48 \times 10^4$ RGE), which correlated with positive results of individual viral-specific PCR tests for BKV and parvovirus B19. For a third patient, we observed a high abundance of human adenovirus B DNA in samples obtained 15 days before ($2.52 \times 10^3$ RGE) and 9 days after ($5.08 \times 10^2$ RGE) diagnosis of adenovirus infection with conventional adenovirus urine PCR. These data

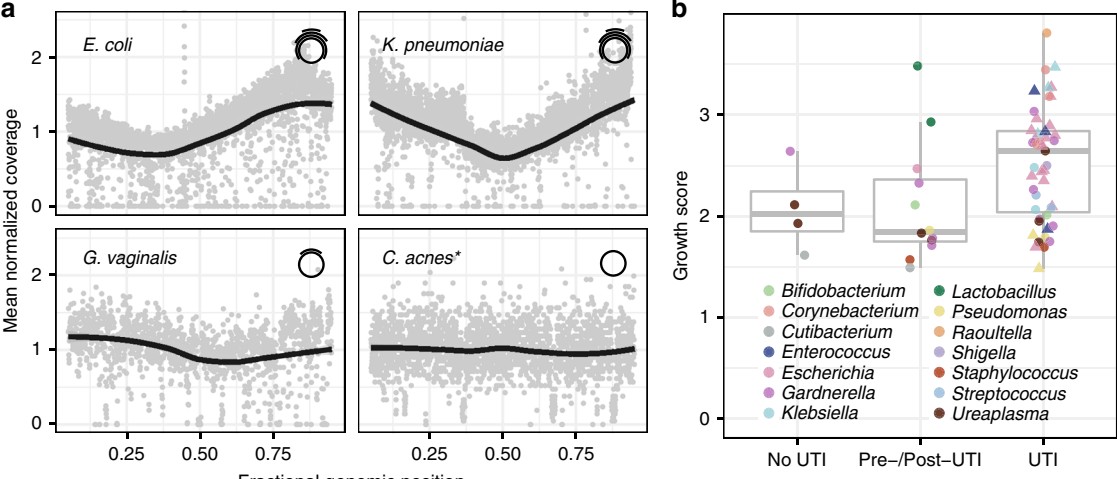

**Fig. 3** Estimating bacterial population growth rates from urinary cfDNA. **a** Normalized bacterial genome coverage for four representative bacterial species. The coverage was binned in 1 kbp tiles and normalized. Each panel represents a single sample (see Supplementary Data 6), with the exception of *C. acnes* (asterisk (*)) for which the coverage was aggregated across 99 samples (solid line is a LOESS filter smoothing curve, span = 0.70). The non-uniform genome coverage for *E. coli* and *K. pneumoniae*, with an overrepresentation of sequences at the origin of replication, is a result of bi-directional replication from a single origin of replication. The initial and final 5% of the genome is removed for display. **b** The skew in genome coverage reflects the bacterial growth rate, where a stronger skew signals faster growth[36]. Box plots of growth rates for species in 14 genera grouped by patient groups (at least 2500 alignments, 41 samples, see Methods for definition of pre/post-UTI). Each point indicates a bacterial species in a sample. Triangles indicate culture-confirmed bacteria by genus. Boxplot features are described in Methods. See Supplementary Data 6 and 7

support the utility of urinary cfDNA sequencing for the detection of both common and uncommon viral agents.

**Quantifying bacterial growth rates.** Conventional metagenomic sequencing can provide a snapshot of the microbiome yet does not inform about microbial life cycles or growth dynamics. In a recent study, Korem and colleagues reported that the pattern of metagenomic sequencing read coverage across a microbial genome can be used to quantify microbial genome replication rates for microbes in complex communities[33]. We tested whether this concept can be used to estimate bacterial population growth from measurements of cfDNA. Figure 3a shows the urinary cfDNA sequence coverage for four bacterial species, *E. coli*, *K. pneumoniae*, *Gardnerella vaginalis*, and *Cutibacterium acnes*. For two subjects diagnosed with *E. coli* and *K. pneumoniae* UTI (Fig. 3a), the *E. coli* and *K. pneumoniae* genome coverage was highly non-uniform, with an overrepresentation of sequences at the origin of replication and an underrepresentation of sequences at the replication terminus. The shape of the *E. coli* and *K. pneumoniae* genome coverage is a result of bi-directional replication from a single origin of replication. The skew in genome coverage reflects the bacterial population growth rate, where a stronger skew signals faster population growth[34]. The genome coverage of a common inhabitant and sometimes uropathogenic bacterial species, *G. vaginalis*, exhibited non-uniform genome coverage (Fig. 3a), similar to the *E. coli* and *K. pneumoniae* cases above but less pronounced. *C. acnes* has been recognized as a common skin commensal and contaminant in the setting of molecular assays[35]. The genome coverage for *C. acnes* was highly uniform, indicative of slow or no growth (aggregate across 99 samples in which *C. acnes* cfDNA was detected, Fig. 3a).

We asked whether this measure of bacterial growth can be used to inform bacterial UTI diagnosis. We calculated an index of replication based on the shape of the sequencing coverage using methods described previously[34]. We used BLAST to identify abundant bacterial strains and then re-aligned all sequences with BWA[36] to a curated list of bacterial species. Samples for which the

genome coverage was too sparse were excluded from this analysis (see Methods). Figure 3b compares the index of replication for bacteria detected by cfDNA in samples from subjects diagnosed with UTI to the index of replication for bacteria detected by cfDNA in samples from subjects with negative cultures and in samples collected from subjects prior to UTI development (Pre-UTI group) or after UTI development (Post-UTI group). Species categorized in the UTI group had markedly greater growth rates than those in the no UTI and Pre-/Post-UTI groups (two-tailed Wilcox rank-sum test, $p = 9.0 \times 10^{-3}$).

**Antimicrobial resistome profiling.** For 42 of the 43 samples collected from subjects with clinically confirmed UTIs, we determined the relative abundance of genes conferring resistance to several classes of antimicrobials (a single sample, for which no AR gene fragments were observed, was excluded from this analysis). We used blastp to align non-human sequences against known AR genes and mutations[37]. AR gene sequences were aggregated and called against the non-redundant Comprehensive Antibiotic Resistance Database that indicates the drug resistance conferred by the given gene[37].

We compared the results of phenotypic antimicrobial susceptibility testing (see Methods) to the resistance profiles determined by cfDNA sequencing. For most samples, there was a high diversity in alignments with highly abundant resistance classes including resistance to macrolides, aminoglycosides, and beta-lactams (Fig. 4). We studied vancomycin-resistant *Enterococcus* (VRE) infections, which often lead to complications after transplantation[7], in depth. Resistance to vancomycin was clinically assessed via measurement of the minimum inhibitory concentration value using broth microdilution on the MicroScan WalkAway platform according to the manufacturer's instructions (see Methods). We detected fragments of genes conferring resistance to the glycopeptide antibiotic class, of which vancomycin is a member, for all VRE-positive samples ($n = 4$). Moreover, for samples with *Enterococcus* that tested as vancomycin susceptible ($n = 7$), we did not detect fragments of

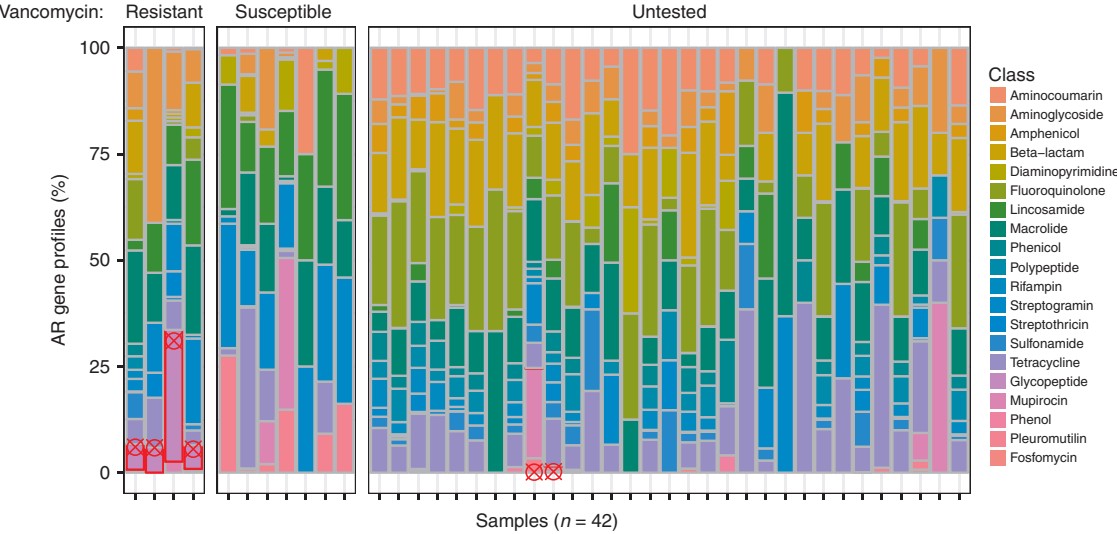

**Fig. 4** cfDNA-based antimicrobial resistome profiling. For 42 samples from subjects with clinically confirmed UTI, AR gene profiling reveals the presence of genes conferring resistance to various antimicrobial classes. These data are organized in three sample groups: samples from subjects with vancomycin-resistant *Enterococcus* (Resistant), samples from subjects with vancomycin-susceptible *Enterococcus* (Susceptible), and samples from subjects for which vancomycin resistance testing was not performed (Untested). Samples in which fragments of genes that confer resistance to glycopeptide class antibiotics (including vancomycin, red outlines) were detected are marked by red crosshairs. See Supplementary Data 8

glycopeptide class resistance genes (Fig. 4). These data indicate potential to predict antimicrobial susceptibility from measurements of urinary cfDNA.

**Host response to infection**. We next examined the host response to viral and bacterial infections. Recent work has identified transplant donor-specific cfDNA in plasma as a marker of graft injury in heart, lung, liver, and kidney transplantation[16,38–40]. Here we quantified donor-specific cfDNA in urine for sex-mismatched donor recipient pairs (i.e., male donor, female recipient; female donor, male recipient) by counting Y chromosome-derived cfDNA (Fig. 5a, Methods). We observed elevated levels of donor cfDNA in the urine of subjects diagnosed with BKVN (mean proportion of donor DNA 65.1%, $n = 12$) compared to the urine of subjects who had normal biopsies (no BKVN, mean 51.4%, $n = 4$) and samples from subjects who did not develop a clinical UTI in the first 3 months of transplantation (mean 25.5%, $n = 11$, samples collected within 5 days after transplant excluded). The release of donor DNA reflects severe cellular and tissue injury in the graft, a hallmark of BKVN. In contrast to subjects with BKVN, subjects diagnosed with bacterial UTI had lower proportions of donor DNA as compared to individuals without bacterial UTI. This is likely explained by an elevated number of recipient immune cells in the urinary tract following immune activation. Indeed, comparison to clinical urinalysis indicates that the donor fraction decreases with increasing white blood cell count (WBC) per high power field (HPF) 400× microscope magnification, inset Fig. 5a, Spearman's rho $= -0.57$, $p = 1.3 \times 10^{-4}$). Furthermore, clinical cases of pyuria, defined as >10 WBC per HPF[41], had a lower donor fraction than those without (two-tailed Wilcox test, $p = 8.0 \times 10^{-4}$). In addition, we found that the level of donor DNA in the first few days after transplant was elevated, consistent with early graft injury. We tracked the relative and absolute abundance of donor-specific urinary cfDNA in the first few days after transplantation for a small subset of subjects ($n = 5$). The initial elevated level of donor cfDNA quickly decayed to a lower baseline level (Fig. 5b), in line with previous observations in heart and lung transplantation[16,42].

Two studies recently demonstrated that the structure of chromatin in gene promoters is conserved within circulating

cfDNA in plasma[43,44]. Ulz et al. employed whole-genome sequencing of plasma DNA to show that nucleosomal occupancy at transcription start sites results in different read depth coverage patterns for expressed and silent genes[44]. Here we found that footprints of nucleosomes in gene promoters and transcriptional regulatory elements are conserved within urinary cfDNA (Fig. 5c, aggregation and normalization across all samples) and that the extent of nucleosomal protection is proportional to gene expression. Measurements of nucleosomal depletion can serve as a proxy for increased gene expression and may be used to investigate host–pathogen interactions in the setting of UTI in more detail.

Mitochondrial DNA (mtDNA) in the urine was recently identified as a possible biomarker for hypertensive kidney damage[45]. Furthermore, recent data indicate a role for extracellular mtDNA as a powerful damage-associated molecular pattern (DAMP). Elevated levels of mtDNA in plasma have been reported in trauma, sepsis, and cancer, and recent studies have identified mtDNA released into the circulation by necrotic cells[46]. For a small subset of subjects diagnosed with BKVN (8 samples from 7 subjects), we quantified donor- and recipient-specific mtDNA in urine using an approach we have previously described[20]. We found that the graft is the predominant source of mitochondrial urinary cfDNA in seven of the eight samples (two-tailed Student's $t$-test, $p < 10^{-6}$; see Methods). Molecular techniques to track DAMPs in urine released in the setting of kidney graft injury may provide a non-invasive window into the potential role of these molecules in immune-related complications.

## Discussion

We have presented a strategy to identify and assess infections of the urinary tract based on profiling of urinary cfDNA and 'omics analysis principles. We show that different layers of clinical information are accessible from a single assay that are either inaccessible using current diagnostic protocols or require parallel implementation of a multitude of different tests. In nearly all samples with clinically reported viral or bacterial infection of the urinary tract, cfDNA identified the suspected causative agent of infection. In addition, cfDNA sequencing revealed the frequent

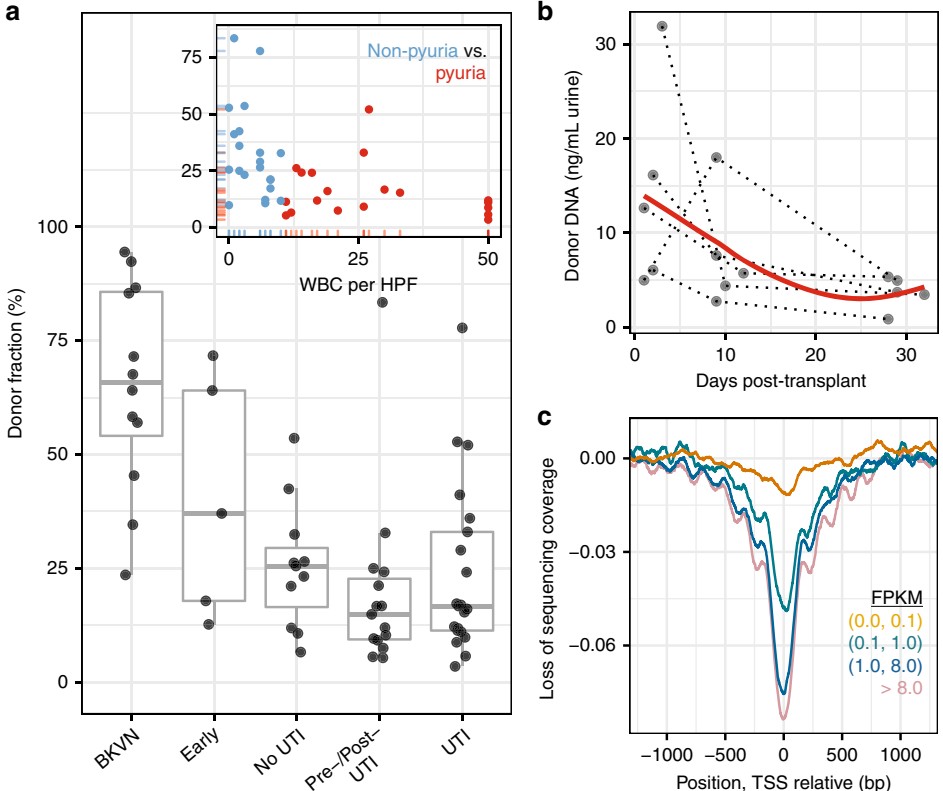

**Fig. 5** Quantifying the host response to infection from urinary cfDNA. **a** Proportion of donor-specific cfDNA in urine of subjects that are BKVN positive per kidney allograft biopsy (BKVN) in urine collected in the first 5 days after transplant surgery (Early), urine collected from subjects that are bacterial UTI negative per culture in the first month following transplantation (No UTI), samples collected before or after bacterial UTI (Pre-/Post-UTI), and samples collected at the time of bacterial UTI diagnosis (UTI). The single outliers in Pre-/Post-UTI and UTI groups correspond to the same patient, who suffered an acute rejection episode in the months prior. Low donor fractions in the Pre-/Post-UTI and UTI groups are likely due to increased immune cell, i.e., WBC, presence in the urinary tract; subjects with higher WBC counts have lower donor fractions (inset, red color indicates pyuria). **b** Absolute abundance of donor cfDNA in the urine of subjects not diagnosed with infection in the first month post-transplant (red line is a LOESS filter smoothing curve, span = 1). Dotted lines connect samples from the same patient. **c** Genome coverage at the transcription start site, binned by the gene expression level across all samples in the study. FPKM fragments per kilobase of transcript per million mapped reads, an RNA-seq measure of gene expression. See Supplementary Data 9 and 10

occurrence of cfDNA from bacteria that remain undetected in current clinical practice. In many samples, including those from subjects regarded as clinically stable, we detected cfDNA from viruses that may be clinically relevant but not routinely assayed in the screening protocol at our institution. The assay we present has the potential to become a valuable tool to monitor bacteriuria and viruria in kidney transplant cohorts and to ascertain their potential impact on allograft health.

Beyond measurement of the abundance of different components of the microbiome, urinary cfDNA provides a wealth of information about bacterial phenotypes. We show, for the first time, that analyses of the structure of microbial genomes from cfDNA allow estimation of bacterial population growth rates, thereby providing information about dynamics from a single snapshot. We compared the bacterial growth rates in samples with clinically diagnosed UTI to those without diagnosed UTI, and we observed higher growth rates for clinically reported bacteria in subjects diagnosed with UTI. We further show that metagenomic analysis of urinary cfDNA can be used to infer antimicrobial susceptibility. We mined cfDNA sequencing data for AR genes and found a good agreement between the presence of AR genes and in vitro phenotypic antimicrobial susceptibility testing results. cfDNA resistome profiling may have added potential over conventional AR testing methods as these methods typically use one

or a few cultured colonies. cfDNA profiling can potentially capture AR gene fragments from the entire bacterial population, which may be particularly important since cfDNA profiling revealed frequent putative co-infections within the UTI group.

Several new methodologies have been introduced in recent years to characterize the urinary microbiome and to diagnose UTI, including 16S rRNA gene sequencing[31,47,48] and expanded culture techniques[9,49]. These approaches have challenged the clinical dogma that urine from healthy individuals is sterile[30] and have revealed potential deficiencies in the culture protocols that are used in clinical practice today[10,31]. The cfDNA shotgun sequencing assay described here provides a versatile alternative that will be particularly useful for monitoring kidney transplant recipients, given the potential to enable viral and bacterial pathogen detection, AR profiling, and graft injury assessment from a single assay.

More than 15,000 patients receive lifesaving kidney transplants in the United States each year[50]. Viral and bacterial infections of the urinary tract occur frequently in this patient group and often lead to serious complications, including graft loss and death. In the general population, UTI is one of the most frequent medical problems that patients present with[51]. Shotgun DNA sequencing of urinary cfDNA offers a comprehensive window into infections of the urinary tract and could be a valuable diagnostic tool to

monitor and diagnose bacterial and viral infections in kidney transplantation as well as in the general population. The assay we have presented is compatible with a short assay turnaround time (1–2 days) and will benefit from continued technical advances in DNA sequencing that will reduce cost and increase throughput in years to come.

## Methods

**Study cohort and sample collection.** One hundred and forty one urine samples were collected from kidney transplant recipients who received care at New York Presbyterian Hospital–Weill Cornell Medical Center. We assayed urine samples from a total of 82 subjects. We assayed urine samples from a total of 82 subjects. The study was approved by the Weill Cornell Medicine Institutional Review Board (protocols 9402002786, 1207012730, 0710009490). All patients provided written informed consent.

Bacterial group: We included 99 urine samples from 34 subjects who developed bacterial UTI diagnosed within the first 12 months of transplantation and 14 subjects who never developed UTI within the first 3 months of transplantation. For the 34 subjects who developed UTIs, we assayed 43 urine samples corresponding to same day positive urine cultures (UTI group); we assayed 15 urine samples from 15 subjects, collected at least 2–16 days (median 7 days) prior to development of the positive urine cultures (Pre-UTI group), and we assayed 12 urine samples from 9 subjects, collected at least 3–26 days (median 9 days) after development of the positive urine cultures (Post-UTI group) (7 of the 9 subjects were treated with antibiotics). We assayed a total of 29 samples collected within 3 months after transplantation from 14 subjects who never developed UTI in the first 3 months of transplantation.

Viral group: The study further included 25 samples from 23 subjects who had a corresponding positive diagnosis of BKVN by needle biopsy of the kidney allograft (BKVN-positive group, 1 sample was also associated with a positive bacterial urine culture) and 10 samples from 10 subjects who had a normal protocol biopsy and were negative for BKV (BKVN-negative group). Finally, the study analyzed seven samples from three subjects who developed clinically diagnosed rare viral infections, including parvovirus or adenovirus. See also detailed clinical metadata in Supplementary Data 1.

**Conventional bacterial culture, bacterial identification, and antimicrobial susceptibility testing.** Ninety of the 141 samples in the study had a corresponding same day urine culture. Each of these clean-catch midstream culture urine samples was inoculated onto tryptic soy agar with sheep blood (Becton, Dickinson and Company [BD], Franklin Lakes, NJ) and MacConkey agar (BD) using a 1-μL inoculation loop and incubated in ambient air at 35 °C. Four urine samples were reported as mixed bacterial flora and were excluded because of lack of further identification and 86 samples were defined as either negative urine culture (n = 43) or positive urine culture (n = 43) for the cfDNA/bacterial correlation analyses and ROC analyses.

A positive urine culture was defined as a culture growing an organism identified to at least the genus level (almost all bacterial isolates were recovered at a colony count ≥10,000 cfu/mL, while three isolates were recovered at a colony count <10,000 cfu/mL). A urine culture was defined as negative when either no organism was isolated in culture (35 cultures, <1000 cfu/mL) or the organism was unidentified to either the genus or species level (i.e., unidentified) and the colony count was <10,000 cfu/mL (8 cultures).

Bacterial isolates were identified using either abbreviated identification algorithms[52] or MicroScan (Beckman Coulter, Inc., West Sacramento, CA) identification panels: Neg ID Type 2 panel for Gram-negative bacteria and Pos Combo 33 panel for Gram-positive bacteria, in conjunction with the WalkAway plus system (Beckman Coulter, Inc.). In two cases, the organism isolated in culture was identified using the MALDI Biotyper CA System (Bruker Daltonics, Inc., Billerica, MA). Testing on the WalkAway plus and MALDI Biotyper CA systems was performed per the manufacturer's instructions.

Antimicrobial susceptibility testing was performed using broth microdilution or disk diffusion. Broth microdilution testing was accomplished using MicroScan antimicrobial susceptibility testing panels on the WalkAway plus system according to the manufacturer's instructions. Gram-negative organisms were tested using the Neg MIC 42 panel or the Pos Combo 33 panel for Gram-positive organisms. In a single instance, an *E. faecalis* isolate was assayed with the Pos MIC 34 panel. An isolate of *H. influenzae* was tested using the disk diffusion method as recommended by the Clinical and Laboratory Standards Institute (CLSI) M02-A12[53]. All antimicrobial susceptibility data were interpreted according to the CLSI M100 document. The M100 version used for interpretation varied depending on the year the isolate was recovered in culture: M100-S25 (2015)[54], M100-S26 (2016)[55], and M100-S27 (2017)[56].

**Conventional viral identification.** Quantitative adenovirus and parvovirus B19 PCR was performed on urine samples in an outside reference laboratory (Viracor Eurofins, Lee's Summit, MO).

**Analysis of discordance against bacterial culture.** In a single sample, urinary cfDNA did not identify the organism reported by conventional culture: *R. ornithinolytica*. The patient had developed an *E. coli* UTI on postoperative day 6 and was treated initially with aztreonam but switched to cephalexin for a 14-day course. The subject subsequently developed a UTI that conventional bacterial culture revealed to be *R. ornithinolytica* on postoperative day 25. cfDNA analysis on urine samples collected on postoperative days 6 and 25 revealed a high abundance of *E. coli* cfDNA and no evidence of *R. ornithinolytica* cfDNA. Given the discordant results, it is unclear if the second culture growing *R. ornithinolytica* is a recurrence as suggested by the cfDNA analysis or is an infection with a different organism as suggested by the urine culture data.

**Urine collection and supernatant isolation.** Most urine samples were collected via the conventional clean-catch midstream culture method (n = 130). Samples obtained prior to post-transplant day 4 were collected via indwelling catheter (n = 11). Approximately 50 mL of urine was centrifuged at 3000 × g on the same day for 30 min and the supernatant was stored at −80 °C in 1 or 4 mL aliquots (except for a single *H. influenzae* UTI sample which was centrifuged for cfDNA analysis 5 days after collection). cfDNA was extracted from 1 mL (131 samples) or 4 mL (10 samples) of urine according to the manufacturer's instructions (Qiagen Circulating Nucleic Acid Kit, Qiagen, Valencia, CA).

**Negative control.** To control for environmental and sample-to-sample contamination, a known-template control sample (IDT-DNA synthetic oligo mix, lengths 25, 40, 55, 70 bp; 0.20 μM eluted in TE buffer) was included with every sample batch and sequenced to a fraction of the depth of the cfDNA extracts (~5 million fragments). The number of bacterial and viral reads detected in the controls was quantified for each genus and normalized to the total reads across the controls. This fractional representation was used to filter out genera detected at low level in the clinical samples: any genus for which the fractional representation in the clinical sample was within five standard deviations from the mean measured in the controls was removed. Possible sources of contamination in these experiments include: environmental contamination during sample collection in the clinic, nucleic acid contamination in reagents used for DNA isolation and library preparation, and sample-to-sample contamination due to Illumina index switching[57].

**Library preparation and next-generation sequencing.** Sequencing libraries were prepared using a single-stranded library preparation optimized for the analysis of ultrashort fragment DNA[20]. Briefly, cfDNA was denatured and biotinylated adapters were ligated via single-stranded DNA ligation. Primer extension was performed on streptavidin functionalized magnetic beads and a second set of adapters was ligated by double-stranded DNA ligation. Finally, the molecules were PCR amplified (4–12 cycles). Libraries were characterized using the AATI fragment analyzer. Samples were pooled and sequenced on the Illumina NextSeq platform (paired-end, 2 × 75 bp). Approximately 45 million paired-end reads were generated per sample.

**Analysis—composition of the urinary microbiome.** Low-quality bases and Illumina-specific sequences were trimmed (Trimmomatic-0.32[58]). Reads from short fragments were merged and a consensus sequence of the overlapping bases were determined using FLASH-1.2.7. Reads were aligned (Bowtie2, very sensitive mode[59]) against the human reference (UCSC hg19). Unaligned reads were extracted, and the non-redundant human genome coverage was calculated (SAMtools 0.1.19 rmdup[60]). To derive the urinary microbiome, reads were BLASTed (NCBI BLAST 2.2.28+) to a curated list of bacterial and viral reference genomes[61]. The relative abundance of different species in a sample was estimated based on the BLAST reports using GRAMMy, a software that implements a maximum likelihood algorithm and takes into account the ambiguity of read mapping[26,38,62]. The relative abundance of higher level taxa was determined based on the relative abundance at the strain or species level. For positive identification of viruses, we required at least 10 BLAST hits. In addition, due to the high load and genetic similarity of BK and JC polyomaviruses, we implemented a conservative filter for incompleteness and heterogeneity of genome coverage (GINI index <0.8 with at least 75% of the genome covered) for these two species only. Code available upon request.

**ROC analysis.** ROC analyses were performed using the function "roc" in the R package pROC. For each species, we compared the relative genomic abundance of species (in RGE) in urine samples matched to a positive culture to the relative genomic abundance of the same species in culture-negative samples.

**Bacterial growth dynamics.** Bacterial genome replication rates were determined using the approach described by Brown et al.[34]. Briefly, all bacterial strains within a sample were sorted and the GC-skew was used to identify the origin and terminus of replication (minimum and maximum GC-skew, respectively). Bacterial genomes were binned in 1 kbp tiles. The coverage was smoothed based on a running mean of 100 nearest neighboring tiles. The coverage in each tile was quantified and tiles were sorted by coverage. Linear regression was performed between the origin and

terminus of replication after further removing the 5% least and most covered bins. The product of the slope of the regression line and the genome length was defined as the growth rate, a metric applied in previous analyses[34]. This analysis was applied for all bacterial strains with genome lengths >0.5 Mbp, $R^2$ linear regression correlation, as previously described, >0.90, and GINI index coefficient <0.2, for which at least 2500 BLAST hits were detected in the sample.

**Nucleosome footprints in gene bodies.** Paired-end reads were aligned using BWA-mem[36]. The sequence read coverage in 2-kbp windows around the transcription start sites of all genes was determined using the SAMtools depth function[44]. A list of transcription start sites organized by transcriptional activity was obtained from Ulz et al.[44]. The depth of coverage was summed across genes with similar transcriptional activity.

**Proportion of donor-specific cfDNA in urine.** The fraction of donor-specific cfDNA in urine and plasma samples was estimated for sex-mismatched, donor–recipient pairs. The donor fraction was determined as follows:

$$\text{Male donor, female recipient}: D = 2Y/A$$

$$\text{Female donor, male recipient}: D = 1 - \left(\frac{2Y}{A}\right)$$

where $Y$ and $A$ are the coverage of the mappability-adjusted Y and autosomal chromosomes, respectively. Sequence mappability was determined using HMMcopy[63].

**Mitochondrial donor fraction.** The proportion of donor-specific mtDNA was quantified using methods previously described[20]. Briefly, mtDNA was extracted from pretransplant whole blood samples, amplified, underwent library preparation, and was sequenced. Processing was separate for donor and recipient samples. Raw sequencing data was trimmed and aligned to the human genome, and mitochondrial sequences were selected. We estimated the presence of each nucleotide at each position using bam-readcount (https://github.com/genome/bam-readcount), and a mitochondrial consensus sequence was determined for the donor and recipient. The consensus sequences were compared and single-nucleotide polymorphisms (SNPs) discriminating the two individuals were identified. Downstream cfDNA sequence data aligning to the mitochondrial were compared to the bases with SNPs that discriminate the donor and recipient. For each SNP position across the mitochondrial genome, we determined the donor fraction by dividing the donor SNP count by the total number of donor and recipient counts at the SNP. We discarded a donor fraction estimation at a point if the depth of sequencing was <50×. We determined the mean of the estimated donor fraction at all SNPs to quantify the mitochondrial donor fraction. One sample was removed owing to low depth of sequencing across all SNPs.

**Antimicrobial resistance profiling.** Paired-end nonhuman sequencing reads were merged (FLASH-1.2.7). Subsequently, reads were aligned to a database of protein sequences, in fasta format, of known AR genes (CARD 1.1.5, 2158 genes) using blastx[64] (evalue $10^{-7}$, culling limit 8 blastx hits). We implemented a filter post-alignment that eliminated reads with <90% similarity between the query and reference. Hits with the highest identity and overlap length were selected for each read. CARD data includes an ontology for the AR class to which each gene confers resistance, if known[37]. We matched the ontology-derived antimicrobial classes described in Fig. 4 to the gene alignment from blastx, giving a measure of each antimicrobial class for each gene hit. If a gene conferred resistance to multiple antimicrobial classes, each class was attributed a hit. The hits were aggregated to provide an antimicrobial susceptibility profile of the sample.

**Statistical analysis.** All statistical analyses were performed using R version 3.3.2. Unless otherwise noted, groups were compared using the nonparametric Mann–Whitney U test. Fourier analyses were performed using the spec.pgram function, part of the standard stats package, in R.

**Boxplots.** Boxes in the boxplots indicate the 25th and 75th percentiles, the band in the box indicates the median, lower whiskers extend from the hinge to the smallest value at most 1.5× IQR of the hinge, and higher whiskers extend from the hinge to the highest value at most 1.5× IQR of the hinge.

**Data availability.** The sequencing data that support the findings of this study are made available in the database of Genotypes and Phenotypes (dbGaP), accession number phs001564.v1.p1.

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

## Acknowledgements

This work was supported by R21AI133331 (to I.D.V. and J.R.L.), R21AI124237 (to I.D.V.), DP2AI138242 (to I.D.V.), K23AI124464 (to J.R.L.), R37AI051652 (to M.S.), and the Robert Noyce Foundation (to I.D.V.). P.B. is supported by an NSF GRFP, DGE-1144153. We thank Erin Berthelsen for providing samples for assay development and Catherine Snopkowski and Carol Li for help with sample handling and qPCR experiments.

## Author contributions

P.B., J.R.L, D.D., M.S., and I.D.V. contributed to the study design. P.B., M.H., and F.C. performed the experiments. P.B., J.R.L., D.D., L.F.W., and I.D.V. analyzed the data. P.B., D.D., L.F.W., J.R.L., and I.D.V. wrote the manuscript. All authors provided comments and edits.

## Additional information

**Competing interests:** The authors declare no competing interests.

