## [Peer Review File · Nature Communications]

Reviewers' comments:

Reviewer #1 (Remarks to the Author):

In this submission, the authors sequence the short fragments of cell-free DNA present in urine collected from kidney transplant patients in an attempt to develop a rapid and comprehensive diagnostic tool. I agree with their assessment that "The assay we present therefore has the potential to become a valuable tool for the monitoring of bacterial and/or viral infections in transplant cohorts, and ascertain their potential impact on allograft health." However, the manuscript contains several flaws (in addition to lacking page and line numbers).

1. Methodology.

A. There is no mention of the sex of the subjects. There is mention of male/female in terms of donor and recipient, but no explicit statement of the sex of the subjects. When analyzing urine, this matters. See next concern.

B. There is no mention anywhere of the urine collection method. From my experience, I assume that the urine was collected the conventional mid-stream void method, but I cannot be sure.

C. If these were voided urines, then the sex of the subject matters. For women, voided urine typically does not truly represent the urinary microbiome, but rather a mixture of urine and vulvo-vaginal microbiome. In most cases, based upon measurements of bacterial biomass in voided urines versus catheterized urines, voided urine is mostly vulvo-vaginal, as this niche has considerably more microbial biomass than the bladder. For men, it is less problematic. I understand the value of voided urine as a feasible method, but conclusions must be adjusted. For accuracy, may I suggest the term genitourinary cfDNA instead of urinary cfDNA.

2. Adherence to the dogma.

A. Over the past 8 years, multiple publications have reported that the urine, the lower urinary tract and/or the bladder is not sterile in the absence of a clinical urinary tract infection. These reports overturn the old "urine is sterile" dogma and lay the foundation for efforts to transform clinical diagnosis and care. The authors must be aware of this state of affairs as they reference one of those papers (Price et al., 2016). Yet, they appear to operate from the basis of the old dogma. They do not expose their readers to the new literature. They compare their results to the conventional urine culture, which has been shown repeatedly to perform poorly except in the detection of the fast growing uropathogens, specifically *E. coli* – no surprise here as that was what the assay was designed to detect. They refer to infections, with little mention of resident microbiota and no mention of dysbiosis. There were lots of reviews in 2015-2017. The authors should read them.

B. Here are a few examples of the problem.

a. In the discussion, the authors state "...we observed higher growth rates for both clinically-diagnosed and co-infecting bacteria in patients with infection." How do the authors know that these bacteria were infecting? Perhaps they were resident bacteria.

Furthermore, instead of infection, the authors should say symptoms. Under the old dogma and current clinical practice, diagnosis arises from the presence of UTI-like symptoms and a conventional urine culture result that detects a known uropathogen, but there is no evidence that the symptoms are caused by the detected bacterium. That has been assumed. In many cases, the assumption is almost certainly true, but not always.

b. In the introduction, the authors state that "The current gold standard for diagnosis of bacterial UTI is in vitro urine culture." But, we should be taken off that gold standard (again, see Price et al., 2016 and the reviews that I mentioned). Thus, the (or a least a) comparison should not be to the old conventional urine culture method, but to the newer approaches. See below.

c. Later, they say "Although improved culture methods are being investigated (Price et al., 2016), bacterial culture is limited to detection of relatively few cultivable organisms." This is only true of

the conventional urine culture method, which only detects fast growing aerobic bacteria with no special nutrient requirements. Most of the urinary microbiota do not grow under these conditions. It is not true for several other approaches, including the enhanced culture method used by Price and co-workers, which also accounts for slow growing, fastidious, microaerophilic and anaerobic bacteria. Other approaches that outperform conventional urine culture for the detection of microbomes are 16S rRNA gene sequencing and shot-gun genomic sequencing (i.e., a metagenomic sequencing method similar but not identical to the authors methods). If the authors had read just a few of the published reports, they would know this.

d. Thus, the authors should compare their results not only to the conventional urine culture, but also to the other methods mentioned above. How does their approach compare to enhanced culture, 16S sequencing and metagenomics in terms of the bacteria and other microbes detected? Furthermore, they should also be upfront and recognize that all approaches have strengths and weaknesses. For example, enhanced culture has the ability to identify to the species level, which the reported method apparently cannot (see below). With the bacterium (whether it is suspected pathogen or suspected commensal), one can sequence its genome or test its antibiotic resistance. However, this one-by-one approach does have limitations, which the reported approach overcomes – for example, it has the advantage of screening the entire community for antibiotic resistance simultaneously.

3. A tendency to overstate.

A. They call their method “unbiased sequencing.” No method is unbiased. I suggest a different term, perhaps one that actually describes the method.

B. They say that their method allows them to “perform robust analyses.” In the Introduction, this appears as simply their opinion with no evidence to back it up. It is unnecessary.

C. In the introduction, they state that “A recent study reports that almost all women with symptoms of UTI but a negative culture still have an infection (Heytens et al., 2017).” This is a mis(over)interpretation by the authors of the cited study. The authors of this study should not follow their example. What the authors of both studies have done is detect bacteria and other microbes in urine of patients with symptoms. In many cases, they have no idea whether these bacteria are invaders or residents. They do not know whether the symptoms are caused by an infection or a dysbiosis. This is another case of not integrating knowledge from the literature.

D. In the results, second paragraph: “These data demonstrate that analyses of the structure of cfDNA can be used to learn about the pathobiology of uropathogens.”

From what I see, the data presented in this section demonstrate that one can get structural data from the sequence reads. I am not sure of the connection to pathobiology of uropathogens? In fact, I am not sure what the authors mean precisely by the term pathobiology? It is a rather broad term. I would not use it so freely. Smacks of overstatement.

E. Infectome screening, last paragraph: “Whereas reports of bacterial culture are skewed towards species that are responsive to culture, cfDNA sequence analyses are sensitive to the full spectrum of uropathogens.” The authors have not shown any evidence that they can detect the full spectrum of uropathogens. They don’t even list the names of the microbes that they consider to be pathogens. Furthermore, as stated above, the authors are comparing their results to a previously demonstrated poor assay. So, cfDNA does well against a low bar. Again, how does it perform relative to enhanced culture methods or 16S sequencing?

F. Cell free DNA. “These data illustrate the disconnect that exists between the frequency of current clinical infection testing and the incidence of viral pathogens in this cohort of transplant patients.” The statement is too strong. Are these all ‘infections?’ Indeed, how does one define infection? Whereas the authors detected these viruses, they made no attempt to compare to non-patient

controls. Are any of these viruses commensals? I posit that the authors have detected viruses. Whether they are infections is untested.

G. Antimicrobial Resistome. "These data indicate significant potential to predict antimicrobial susceptibility from measurements of urinary cfDNA."

Here is a well-measured conclusion. The authors should generalize this approach.

Other comments/concerns.

1. Infectome screening, last paragraph: : "Whereas reports of bacterial culture are skewed towards species that are responsive to culture, cfDNA sequence analyses are sensitive to the full spectrum of uropathogens." Did the authors detect "commensals" or emerging pathogens? Detection of the listed bacterial genera is easy. Also, were the authors able to identify at the species level? This matters. There are pathogens and non-pathogens in many genera. Indeed, there are both in many species. Treatment of a non-pathogen would not be a good idea.

2. "In addition, bacterial culture is unable to inform about commensal microbiota, viral infections, or about bacterial growth dynamics." The first statement is patently untrue. Even conventional culture methods can detect some commensals. They just are not reported. In this regard, the authors should talk to a clinical microbiologist. Furthermore, enhanced urine culture techniques can detect lots of microbes (hundreds of species). Most are probably commensals and not pathogens. I say probably because the jury is still out. In any case, beyond a few genera that are mentioned, the authors do not actually demonstrate ability to detect commensals. If they can detect them, they should show the data and (again) compare to published reports that used more sensitive techniques than conventional urine culture.

3. Quantifying growth rates. "Species categorized in the UTI group..."

The figure shows genus not species. It is also very difficult if not impossible to distinguish some genera from others with a very similar color. Furthermore, the UTI cohort could be divided into two clusters - one with a high index and one with a low index. So, yes, one can determine which bacteria are growing faster and some of these fast growing bacteria are likely pathogens. But, this plays into the current dogma - that only fast growing bacteria, like E. coli and Pseudomonas, cause UTI symptoms. There is published data that suggests that this is not always the case. There are slow growing Gram-positive bacteria that are thought to be emerging pathogens and which appear to be associated with urinary tract infections. Again, read the new literature carefully.

4. Quantifying growth rates. "...growth from cfDNA may enable identification of virulent microbial strains and evaluation of the response to anti-bacterial drug treatments."

In terms of the former state, yes, but only if virulence is associated with rapid growth, which may not always be the case. The latter statement does not follow from the data above, unless the authors wish to inform the reader that many antibiotics (e.g. beta-lactams) only work well on rapidly growing bacteria.

5. Discussion. "cfDNA identified the causative agent of infection."

For correctly, the authors should say "suspected causative agent." How does one know that the culture bacterium was the causative agent? Detection of a bacterium or virus that can be a pathogen does not confirm etiology. For example, there are strains of E. coli that are clearly pathogens, including some that can kill in a matter of hours. Other strains are perfectly harmless. In fact, some are used as probiotics. Be careful.

6. Analysis of discordance. "The subject had developed a prior Escherichia coli UTI by conventional bacterial culture..." LOL. The subject did not develop a UTI by conventional culture. What the authors mean is "develop a UTI as detected by conventional culture."

7. Negative culture. "...the 23 microbiome controls consisted..."

I do not understand. What are these controls? And why did contain high percentages of typical

pathogens?

8. Results, first paragraph: "...microbiota, for example..." – this should be a semicolon. The sentence is very confusing with a comma.

9. Infectome Screening, second paragraph: "This high load of BK derived DNA is consistent with the pathobiology of BKVN."

Isn't this just a restatement of the previous sentence. It's kind of a 'duh' summary and smacks of an attempt to overstate the authors' case.

10. Figure 2d legend. What do the colors of the dots represent?

11. In several places, the authors refer to supplemental data tables. I only see one untitled supplemental data table.

Reviewer #2 (Remarks to the Author):

In this manuscript, the authors describe the use of cell-free DNA collected from urinary specimens to simultaneously assay bacterial, viral, and host information in a cohort of kidney transplant recipients. Both urinary tract and viral infections represent common complications in kidney transplant recipients, and the results of this study suggest that the bulk sequencing of cell-free urinary DNA may provide a useful strategy for pathogen monitoring, antibiotic resistance profiling, and patient monitoring in the context graft tissue injury.

Although the approach described and results obtained are very interesting and show tremendous promise for highly personalized care, the manuscript would benefit from the consideration of the following:

1) Throughout the methods section, paragraphs are mismatched with respect to verb tense and presentation. I would encourage the authors to provide a more uniform presentation throughout this section of the manuscript.

2) The methods section is also lacking key details that would preclude a reader from replicating this work. For example, 1) the analysis of plasma samples is described in the paragraph headlined with "Proportion of donor-specific cfDNA in urine", but blood collection is not included in the methods section; 2) The paragraph describing antimicrobial susceptibility testing does not include which antimicrobials were tested or under what (specific) conditions (e.g. growth media, temperatures, kits, etc.); 3) Fourier analysis is described in the legend for Figure 1 but not explicitly described in the methods.

3) The figure legends would benefit from the inclusion of additional information. For example, in Figure 1 C, it would be helpful to know how many samples were included in the calculation of each ROC, and a note explaining that the gray color indicates the overlap of the red/blue-green colors would be helpful to the reader. In Figure 1D, please provide additional clarification regarding the meaning of pre- and post UTI. In Figure 3B, what are the values represented by the box and whisker plots? Are these medians and interquartile ranges, means and standard deviations, etc? In Figure 4, were these culture confirmed, and if so, to what degree did the cfDNA-based results match the culture-based results? In figure 5B, please provide additional information regarding the individual vs. aggregate lines and the error bars associated with each of the individual lines.

4) While I applaud the authors' intent to deposit their data with NCBI, full deposition information would be preferable.

5) Although the authors do give a nod to the fact that the clinical feasibility of cfDNA sequencing as a diagnostic assay will likely increase as sequencing costs and turnaround time (TAT) continue to decline, it would be helpful to see a comparison of current TAT for the assay vs current diagnostic assays.

Point-by-point address of the specific comments raised by the reviewers.

(Original report text in italic; our report in blue font)

Reviewer #1 (Technical Comments to the Author):

In this submission, the authors sequence the short fragments of cell-free DNA present in urine collected from kidney transplant patients in an attempt to develop a rapid and comprehensive diagnostic tool. I agree with their assessment that “The assay we present therefore has the potential to become a valuable tool for the monitoring of bacterial and/or viral infections in transplant cohorts, and ascertain their potential impact on allograft health.” However, the manuscript contains several flaws (in addition to lacking page and line numbers).

We thank the reviewer for many detailed comments that have allowed us to improve the data analysis, presentation, and discussion. We are happy the reviewer supports our central premise that the urinary cell-free DNA (cfDNA) assay presented in this work has the potential to become a valuable tool for monitoring viral and bacterial infection of the urinary tract, and to assess their impact on allograft health.

We describe results from significant additional experiments and address the comments of the reviewer point by point below.

1. *Methodology.*

A. *There is no mention of the sex of the subjects. There is mention of male/female in terms of donor and recipient, but no explicit statement of the sex of the subjects. When analyzing urine, this matters. See next concern.*

Information on the sex of the subjects was provided in the supplemental data table “Clinical Data”, column “Recipient Gender” (see “other comments” #11). The reviewer makes an excellent point: we have followed this suggestion and have analyzed the relationships between donor and recipient gender and the cfDNA microbiome. See 1C.

B. *There is no mention anywhere of the urine collection method. From my experience, I assume that the urine was collected the conventional mid-stream void method, but I cannot be sure.*

Most samples were collected via the conventional mid-stream void method. A small number of samples (n = 11) were obtained prior to post-transplant day 4 and were collected via Foley catheter as standard protocol is to decompress the bladder for the first 3 to 4 days after kidney transplantation. We have updated the methods section with this additional information. We have analyzed the relationship between the microbiome and the urine collection method (see 1C).

C. *If these were voided urines, then the sex of the subject matters. For women, voided urine typically does not truly represent the urinary microbiome, but rather a mixture of urine and vulvo-vaginal microbiome. In most cases, based upon measurements of bacterial biomass in voided urines versus catheterized urines, voided urine is mostly vulvo-vaginal, as this niche has considerably more microbial biomass than the bladder. For men, it is less problematic. I understand the value of voided urine as a feasible method, but conclusions must be adjusted. For accuracy, may I suggest the term genitourinary cfDNA instead of urinary cfDNA.*

Our goal for this paper is to introduce shotgun sequencing of urinary cfDNA as a strategy to monitor viral and bacterial infections and graft injury, with an emphasis on the versatility of the assay. We refrained from using urinary cfDNA as a tool to describe the urinary microbiome here. Nevertheless, in the revised version of the paper, we provide additional analyses of the relationships between the microbiome abundance and composition and the recipient gender, the donor gender and the urine collection method. We find that the recipient gender and the urine collection method, but not the donor gender, have a significant effect on the abundance and diversity of the bacterial component of the microbiome, as suggested by the reviewer (see Fig. S2, and the “Profiling the urinary microbiome” section).

2. *Adherence to the dogma.*

A. *Over the past 8 years, multiple publications have reported that the urine, the lower urinary tract and/or the bladder is not sterile in the absence of a clinical urinary tract infection. These reports overturn the old*

“urine is sterile” dogma and lay the foundation for efforts to transform clinical diagnosis and care. The authors must be aware of this state of affairs as they reference one of those papers (Price et al., 2016). Yet, they appear to operate from the basis of the old dogma. They do not expose their readers to the new literature. They compare their results to the conventional urine culture, which has been shown repeatedly to perform poorly except in the detection of the fast growing uropathogens, specifically E. coli – no surprise here as that was what the assay was designed to detect. They refer to infections, with little mention of resident microbiota and no mention of dysbiosis. There were lots of reviews in 2015-2017. The authors should read them.

First, we agree with the reviewer that a deeper discussion of the recent literature is warranted, and we have updated the manuscript with more background in the results and discussion sections.

Second, we show a wide range of analyses made possible through shotgun sequencing of cfDNA, including but not limited to analyses of bacteriuria. To benchmark this particular aspect, we compare cfDNA to conventional bacterial culture (but we do include analyses against broader spectrum bacterial culture, see below), because conventional bacterial culture is standard practice in the vast majority of clinical diagnostic centers (see results from survey below). We report excellent agreement between urinary cfDNA and conventional culture (bacteria reported in culture were detected in urinary cfDNA in 42/43 cases). Furthermore, we performed additional analyses on samples from patients with clinically reported bacterial and viral infections, and these experiments have confirmed and strengthened all our conclusions.

Third, in the new version of the manuscript, we do include analyses of the resident microbiota as function of recipient gender, donor gender, and urine collection method.

Fourth, to address the utility of cfDNA to detect bacterial infectious agents that are not responsive to conventional culture, or viral agents that are not routinely screened for in current diagnostic protocols, we performed several additional experiments:

1. Direct comparison to conventional bacterial culture: we assayed two samples from a patient who was suspected of having *H. influenzae* urosepsis. While the urinalysis was markedly positive (> 50 WBC/HPF), repeated conventional urine cultures were negative. Given the clinical suspicion and given that *H. influenzae* does not routinely grow on tryptic soy agar with sheep blood agar or MacConkey agar, clinicians requested the urine to be plated on chocolate agar, which was positive for *H. influenzae*. In both a sample taken at presentation as well as 4 days after antibiotic treatment, we detected high amounts of *H. influenzae* cfDNA. It is worth noting that *H. influenzae* as a cause of UTI is rarely reported and there are only a few case reports in the transplant literature (Kim et al., Lab Med Online 2(3):170-173, 2012). This case supports the utility of urinary cfDNA sequencing to identify bacteriuria in cases where conventional culture is negative, and provides a direct comparison of cfDNA to a non-standard, expanded spectrum bacterial culture method as requested by the reviewer.

2. Identifying uncommon viral infections: We performed additional analyses for three subjects diagnosed with viral infections that can cause serious complications in our cohort but that are not routinely screened for. In one case of adenovirus viruria, we detected cfDNA in the urine specimen 15 days prior to the clinical diagnosis. In two cases of infection by parvovirus B19, we detected cfDNA in the urine specimen 8 days prior to the clinical diagnosis in one case and 80 days prior to the clinical diagnosis in another case. Therefore, we believe viral cfDNA could be detectable weeks to months earlier than associated symptoms would arise. It is worth noting that cfDNA does detect BK virus as well as cytomegalovirus (Fig. 2D), which are commonly screened for in the kidney transplant population, further supporting the advantages of comprehensive cfDNA sequencing compared to pathogen-specific assays for both common and uncommon viral infections.

We believe that these additional experiments show that cfDNA not only has excellent diagnostic capacity when compared to the current gold standard, but also performs well in scenarios where conventional culture

(i.e., as in the case where we recovered *H. influenzae*) fails, and current pathogen-specific screening protocols (uncommon viral infections) do not detect agents of infection.

Finally, we decided against performing a more systematic comparison to enhanced quantitative urine culture (EQUC, beyond the single case described above), in view of the following reasons: 1) Such analyses would require *de novo* patient recruitment, approval of a new study, and collection of fresh urine (cfDNA testing was performed on frozen, banked urine supernatant). 2) We conducted a survey to understand the current use of EQUC in current clinical practice in the United States. We sent out a survey to clinical microbiology laboratory directors, and learned that only 2 of 35 (5.7%) respondents utilize this method, and then only in isolated cases, and typically only upon request by specialty clinical services (e.g., urology and infectious diseases). Therefore, conventional culture clearly remains the gold standard for diagnosis of bacteriuria and UTI. 3) We do not want to distract from the main message of this paper, namely that urinary cfDNA offers a highly versatile approach for the monitoring of infections of the urinary tract, and is not only limited to detecting bacteriuria.

B. Here are a few examples of the problem.

a. In the discussion, the authors state "...we observed higher growth rates for both clinically-diagnosed and co-infecting bacteria in patients with infection." How do the authors know that these bacteria were infecting? Perhaps they were resident bacteria.

Furthermore, instead of infection, the authors should say symptoms. Under the old dogma and current clinical practice, diagnosis arises from the presence of UTI-like symptoms and a conventional urine culture result that detects a known uropathogen, but there is no evidence that the symptoms are caused by the detected bacterium. That has been assumed. In many cases, the assumption is almost certainly true, but not always.

We have dropped this sentence. We appreciate the nuances indicated by the reviewer. As the reviewer points out, diagnosis of infection remains complex.

b. In the introduction, the authors state that "The current gold standard for diagnosis of bacterial UTI is in vitro urine culture." But, we should be taken off that gold standard (again, see Price et al., 2016 and the reviews that I mentioned). Thus, the (or a least a) comparison should not be to the old conventional urine culture method, but to the newer approaches. See below.

We have performed additional experiments to include a comparison to an expanded culture as requested by the reviewer (response 2A). These additional experiments highlight that cfDNA not only has excellent diagnostic capacity when compared to the gold standard of conventional urine culture but also can detect bacterial infection where conventional urine culture is falsely negative, and furthermore enables prediction of the development of viral infections prior to clinical diagnosis of viremia.

In addition, as noted above, we surveyed clinical microbiology laboratory directors in the US and learned that only 2 of 35 (5.7 %) respondents employ expanded culture for bacterial uropathogens in any capacity, and typically only when requested by specialty clinical services (i.e, it is not an orderable test routinely available to physicians, but must be requested by physicians consulting on specific clinical services, such as urology and infectious diseases).

c. Later, they say "Although improved culture methods are being investigated (Price et al., 2016), bacterial culture is limited to detection of relatively few cultivable organisms." This is only true of the conventional urine culture method, which only detects fast growing aerobic bacteria with no special nutrient requirements. Most of the urinary microbiota do not grow under these conditions. It is not true for several other approaches, including the enhanced culture method used by Price and co-workers, which also accounts for slow growing, fastidious, microaerophilic and anaerobic bacteria. Other approaches that outperform conventional urine culture for the detection of microbomes are 16S rRNA gene sequencing and shot-gun genomic sequencing (i.e., a metagenomic sequencing method similar but not identical to the authors methods). If the authors had read just a few of the published reports, they would know this.

We rephrase to “Although improved culture methods are being investigated (Price et al., 2016), conventional bacterial culture is limited to detection of relatively few cultivable organisms.” We furthermore refer to recent 16S rRNA gene sequencing studies in the discussion section.

d. Thus, the authors should compare their results not only to the conventional urine culture, but also to the other methods mentioned above. How does their approach compare to enhanced culture, 16S sequencing and metagenomics in terms of the bacteria and other microbes detected? Furthermore, they should also be upfront and recognize that all approaches have strengths and weaknesses. For example, enhanced culture has the ability to identify to the species level, which the reported method apparently cannot (see below). With the bacterium (whether it is suspected pathogen or suspected commensal), one can sequence its genome or test its antibiotic resistance. However, this one-by-one approach does have limitations, which the reported approach overcomes – for example, it has the advantage of screening the entire community for antibiotic resistance simultaneously.

First, we have performed additional experiments and assess cfDNA to expanded culture methods for a specific case (*H. influenzae* UTI). We have also performed additional assays on samples from patients diagnosed with less common viral infections, and show significant utility in the early and sensitive detection in such scenarios where diagnosis is more challenging with conventional pathogen-specific diagnostic workups.

Second, we have updated the manuscript with additional discussion of methodologies introduced recently (16S rRNA gene deep sequencing, and enhanced quantitative urine culture).

Third, the cfDNA shotgun sequencing approach is inherently compatible with species level identification. For patients that develop bacterial UTI, a high per-base coverage of bacterial genomes is achieved (often 10x, in some cases > 1,000x coverage of the full bacterial genome, meaning that every position in the genome is measured > 1,000 times). The original version of Fig. 2C reported genus level comparisons. This was an unfortunate choice. We have changed this figure and now show analyses of the performance to identify specific species. We also note that we have performed significant additional experiments to evaluate the performance to assess bacterial UTI (15 additional sequencing assays). These additional experiments further strengthen all the statistical comparisons, and further provide support for our main premise.

3. A tendency to overstate.

A. They call their method “unbiased sequencing.” No method is unbiased. I suggest a different term, perhaps one that actually describes the method.

The term unbiased sequencing is used in biomedical genomics to distinguish whole genome assays from targeted, PCR-based sequencing strategies. We agree that the term may lead to confusion in the context of the evaluation of a clinical test. Therefore, we dropped the term and instead write “shotgun sequencing.”

B. They say that their method allows them to “perform robust analyses.” In the Introduction, this appears as simply their opinion with no evidence to back it up. It is unnecessary.

We wanted to highlight consistent success in assaying urinary cfDNA isolated from relatively small volumes of urine (which is non-trivial given the highly fragmented nature of cfDNA in urine compared to plasma). We have followed the advice of the reviewer, and have removed the term “robust.”

C. In the introduction, they state that “A recent study reports that almost all women with symptoms of UTI but a negative culture still have an infection (Heytens et al., 2017).” This is a mis(over)interpretation by the authors of the cited study. The authors of this study should not follow their example. What the authors of both studies have done is detect bacteria and other microbes in urine of patients with symptoms. In many cases, they have no idea whether these bacteria are invaders or residents. They do not know whether the symptoms are caused by an infection or a dysbiosis. This is another case of not integrating knowledge from the literature.

We agree that this is not the best study to reference in the introduction section. We have reworked the introduction to better integrate recent knowledge.

D. In the results, second paragraph: “These data demonstrate that analyses of the structure of cfDNA can be used to learn about the pathobiology of uropathogens.”

From what I see, the data presented in this section demonstrate that one can get structural data from the sequence reads. I am not sure of the connection to pathobiology of uropathogens? In fact, I am not sure what the authors mean precisely by the term pathobiology? It is a rather broad term. I would not use it so freely. Smacks of overstatement.

The cell-free DNA fragment length profile for BK virus reveals that the DNA is protected and stabilized by histones, revealing some of the known biology for this virus. Nevertheless, we agree that using the term pathobiology is unnecessary here, and have removed it in the updated manuscript.

E. Infectome screening, last paragraph: “Whereas reports of bacterial culture are skewed towards species that are responsive to culture, cfDNA sequence analyses are sensitive to the full spectrum of uropathogens.” The authors have not shown any evidence that they can detect the full spectrum of uropathogens. They don’t even list the names of the microbes that they consider to be pathogens. Furthermore, as stated above, the authors are comparing their results to a previously demonstrated poor assay. So, cfDNA does well against a low bar. Again, how does it perform relative to enhanced culture methods or 16S sequencing?

First, we have altered this sentence to: “Whereas bacterial culture is skewed towards species that are readily isolated on routine bacteriological media employed for urine culture, cfDNA sequence analyses potentially permit the identification of a broader spectrum of bacterial species.” We have removed the term uropathogens as the reviewer correctly points out that we do not evaluate whether the bacteria are pathogenic.

Second, we have performed additional experiments and directly assess cfDNA to a form of expanded culture (see also response to 2A, 2Bd).

F. Cell free DNA. “These data illustrate the disconnect that exists between the frequency of current clinical infection testing and the incidence of viral pathogens in this cohort of transplant patients.” The statement is too strong. Are these all ‘infections?’ Indeed, how does one define infection? Whereas the authors detected these viruses, they made no attempt to compare to non-patient controls. Are any of these viruses commensals? I posit that the authors have detected viruses. Whether they are infections is untested.

We have dropped this sentence. We provide additional support for the utility of cfDNA for the broad screening for viruses. We assayed serial urine from three patients, 2 parvovirus cases and one adenovirus case (see also 2A), and we detected the viruses via cfDNA prior to clinical diagnosis (up to 80 days) and after clinical diagnosis (up to 25 days). Again, we agree with the reviewer that defining “infection” is complex. In the three cases above, however, the patients were diagnosed with clinically significant infections. In both parvovirus cases, the subjects eventually presented with severe anemia and in the adenovirus case, the subject had severe dysuria, hematuria, and fevers with conventional urine cultures repeatedly negative. To err on the side of caution, we have dropped the sentence quoted by the reviewer.

G. Antimicrobial Resistome. “These data indicate significant potential to predict antimicrobial susceptibility from measurements of urinary cfDNA.”

Here is a well-measured conclusion. The authors should generalize this approach.

We thank the reviewer for the appreciation of the antimicrobial susceptibility analysis. In the new version of the manuscript, we include data for 15 additional UTI cases. These additional experiments support and further strengthen our original conclusions, including those related to resistome profiling.

Other comments/concerns.

1. Infectome screening, last paragraph: : “Whereas reports of bacterial culture are skewed towards species that are responsive to culture, cfDNA sequence analyses are sensitive to the full spectrum of uropathogens.” Did the authors detect “commensals” or emerging pathogens? Detection of the listed bacterial genera is easy. Also, were the authors able to identify at the species level? This matters. There are pathogens and

non-pathogens in many genera. Indeed, there are both in many species. Treatment of a non-pathogen would not be a good idea.

We implement shotgun sequencing in this study. The per-base coverage of bacterial genomes of patients diagnosed with urinary tract infection is in many cases greater than 10x, in some cases greater than 1,000x. Such whole-genome analyses with high per-base coverage enable identification of specific species, and in principle enable characterization of the strains by single nucleotide variants and assessment of specific genes, as is exemplified with the resistome analysis. We include species level analyses in the new version of the manuscript. In figures S2, we provide receiver operating characteristics analysis for a total of six different species and two different genera that were reported in culture. These analyses reveal that cell-free DNA performs well in identifying organisms to the species level.

2. *“In addition, bacterial culture is unable to inform about commensal microbiota, viral infections, or about bacterial growth dynamics.” The first statement is patently untrue. Even conventional culture methods can detect some commensals. They just are not reported. In this regard, the authors should talk to a clinical microbiologist. Furthermore, enhanced urine culture techniques can detect lots of microbes (hundreds of species). Most are probably commensals and not pathogens. I say probably because the jury is still out. In any case, beyond a few genera that are mentioned, the authors do not actually demonstrate ability to detect commensals. If they can detect them, they should show the data and (again) compare to published reports that used more sensitive techniques than conventional urine culture.*

We have dropped this sentence. The analysis of microbiome-gender relationships included in the new version of the text, exemplifies the sensitivity of cfDNA to members of the female vaginal microbiome (*Gardnerella* and *Lactobacillus* for example). This is not surprising given that our assay is based on whole-genome shotgun sequencing.

3. *Quantifying growth rates. “Species categorized in the UTI group....”*

*The figure shows genus not species. It is also very difficult if not impossible to distinguish some genera from others with a very similar color. Furthermore, the UTI cohort could be divided into two clusters - one with a high index and one with a low index. So, yes, one can determine which bacteria are growing faster and some of these fast growing bacteria are likely pathogens. But, this plays into the current dogma - that only fast growing bacteria, like *E. coli* and *Pseudomonas*, cause UTI symptoms. There is published data that suggests that this is not always the case. There are slow growing Gram-positive bacteria that are thought to be emerging pathogens and which appear to be associated with urinary tract infections. Again, read the new literature carefully.*

The individual data points in Fig 3B are growth rate measurements for individual species, the coloring is based on genus. We have clarified this in the text. A supplemental data table was and is provided to summarize the individual measurements (see also Q11).

4. *Quantifying growth rates. “...growth from cfDNA may enable identification of virulent microbial strains and evaluation of the response to anti-bacterial drug treatments.”*

In terms of the former state, yes, but only if virulence is associated with rapid growth, which may not always be the case. The latter statement does not follow from the data above, unless the authors wish to inform the reader that many antibiotics (e.g. beta-lactams) only work well on rapidly growing bacteria.

We have removed this sentence.

5. *Discussion. “cfDNA identified the causative agent of infection.”*

*For correctly, the authors should say “suspected causative agent.” How does one know that the culture bacterium was the causative agent? Detection of a bacterium or virus that can be a pathogen does not confirm etiology. For example, there are strains of *E. coli* that are clearly pathogens, including some that can kill in a matter of hours. Other strains are perfectly harmless. In fact, some are used as probiotics. Be careful.*

This is a good suggestion. We have updated the manuscript, and say “suspected causative agent” where relevant.

6. *Analysis of discordance.* “The subject had developed a prior *Escherichia coli* UTI by conventional bacterial culture....” LOL. The subject did not develop a UTI by conventional culture. What the authors mean is 'develop a UTI as detected by conventional culture.'

We fixed this typo.

7. *Negative culture.* “...the 23 microbiome controls consisted....”

I do not understand. What are these controls? And why did contain high percentages of typical pathogens?

A negative, known-template control was included in all DNA sequencing experiments to control for possible environmental or sample-to-sample contamination, but also artifacts inherent to Illumina sequencing: we multiplex 10-20 samples on an individual Illumina sequencing lane as is almost always done with Illumina sequencing, and consequently we are sensitive to “barcode hopping”, a phenomenon inherent to Illumina sequencing where an incorrect sample barcode is assigned to a very small subset of reads (this also happens for 16S sequencing). This occurs with a frequency of a few parts per million, but can for example lead to detection of a small number of BK polyomavirus sequences in a control sample that is sequenced along with a sample with a very high load of BK DNA. The controls allow to account for these phenomena. The addition of controls is good practice, and should be more widely adopted in studies that utilize DNA sequencing.

8. *Results, first paragraph:* “...microbiota, for example...” – this should be a semicolon. The sentence is very confusing with a comma.

We fixed this typo.

9. *Infectome Screening, second paragraph:* “This high load of BK derived DNA is consistent with the pathobiology of BKVN.”

Isn't this just a restatement of the previous sentence. It's kind of a 'duh' summary and smacks of an attempt to overstate the authors' case.

We have dropped this sentence.

10. *Figure 2d legend.* What do the colors of the dots represent?

Viruses from different orders are represented in this figure with different colors to provide structure, and clarity. We have updated the figure legend to indicate this.

11. *In several places, the authors refer to supplemental data tables. I only see one untitled supplemental data table.*

We had uploaded a single excel sheet, comprising multiple data tables. The individual data tables can be accessed by toggling the tabs at the bottom of the sheet.

Reviewer #2 (Technical Comments to the Author):

In this manuscript, the authors describe the use of cell-free DNA collected from urinary specimens to simultaneously assay bacterial, viral, and host information in a cohort of kidney transplant recipients. Both urinary tract and viral infections represent common complications in kidney transplant recipients, and the results of this study suggest that the bulk sequencing of cell-free urinary DNA may provide a useful strategy for pathogen monitoring, antibiotic resistance profiling, and patient monitoring in the context graft tissue injury. Although the approach described and results obtained are very interesting and show tremendous promise for highly personalized care, the manuscript would benefit from the consideration of the following:

We thank the reviewer for the appreciation of our work and the careful reading of our manuscript.

1) Throughout the methods section, paragraphs are mismatched with respect to verb tense and presentation. I would encourage the authors to provide a more uniform presentation throughout this section of the manuscript.

We thank the reviewer for pointing this out. We have improved the writing of the methods section.

2) *The methods section is also lacking key details that would preclude a reader from replicating this work. For example, 1) the analysis of plasma samples is described in the paragraph headlined with “Proportion of donor-specific cfDNA in urine”, but blood collection is not included in the methods section; 2) The paragraph describing antimicrobial susceptibility testing does not include which antimicrobials were tested or under what (specific) conditions (e.g. growth media, temperatures, kits, etc.); 3) Fourier analysis is described in the legend for Figure 1 but not explicitly described in the methods.*

We provide detailed description of the methodologies used, including antimicrobial susceptibility testing, and the Fourier analysis (methods). We note that all experiments in this study are performed on urine; we did not assay plasma. The fraction of donor cfDNA was determined for urinary cfDNA, not plasma cfDNA. We determined the fraction of donor DNA in urine based on a measurement of the Y chromosome sequence coverage for sex-mismatched patients. We benchmarked this approach against a method based on the analysis of SNP markers that we previously described, and here we have taken advantage of data from a previous study (lung transplant cohort, no additional experiments on plasma required).

3) *The figure legends would benefit from the inclusion of additional information. For example, in Figure 1 C, it would be helpful to know how many samples were included in the calculation of each ROC, and a note explaining that the gray color indicates the overlap of the red/blue-green colors would be helpful to the reader. In Figure 1D, please provide additional clarification regarding the meaning of pre- and post UTI. In Figure 3B, what are the values represented by the box and whisker plots? Are these medians and interquartile ranges, means and standard deviations, etc? In Figure 4, were these culture confirmed, and if so, to what degree did the cfDNA-based results match the culture-based results? In figure 5B, please provide additional information regarding the individual vs. aggregate lines and the error bars associated with each of the individual lines.*

We have updated the figure legends with the additional information (Fig. 1C, 1D, and 5B). We provide details on the interpretation of the boxplots in the methods. A new version of figure 4 is shown that shows excellent agreement with vancomycin resistance screening. We provide individual ROC analyses in the supplement. We note that we have performed significant additional experiments on urine matched to positive conventional culture. The results of these additional analyses confirm and strengthen our initial conclusions. In Figure 5B, we have defined the individual and aggregate lines in the legend. We have removed the error bars from each of the lines.

4) *While I applaud the authors’ intent to deposit their data with NCBI, full deposition information would be preferable.*

We will make all sequence data, and matching clinical data available in the database of Genotypes and Phenotypes (dbGaP). We are working with our Program Officer at NIH to finalize registration of the study with dbGaP.

5) *Although the authors do give a nod to the fact that the clinical feasibility of cfDNA sequencing as a diagnostic assay will likely increase as sequencing costs and turnaround time (TAT) continue to decline, it would be helpful to see a comparison of current TAT for the assay vs current diagnostic assays.*

The run time for the Illumina NextSeq550 platform is on the order of 15 hours (2x75 bp, Mid-Output kit). A novel sequencing platform introduced by Illumina (iSeq100) has the potential to enable a sample turnaround from DNA to sequence data under 12h. The turnaround time for sequencing assays already compares favorably to culture-based approaches, and we expect new technologies will provide further improvements on this front in the near term.

Reviewers' comments:

Reviewer #1 (Remarks to the Author):

The authors have been responsive to the reviewers and the result is a much better manuscript. I have only 3 suggestions.

Page 4, line 33. "...fewer DNA fragments that are shorter" is ambiguous. Do you mean fewer shorter DNA fragments with no effect on longer fragments? I think not. It would be crystal clear if it read "...fewer DNA fragments that are [also] shorter. "

Page 8. Line 31. "...typically commensal bacterial species, *G. vaginalis*..." I would be careful about calling *G. vaginalis* a commensal. In the vagina, it's a bit controversial; some studies find it associated with bacterial vaginosis, while others do not. In the adult female bladder (transurethral catheter-collected urines), it is associated with urgency urinary incontinence (Pearce et al., 2014). I would back off and call it a common inhabitant of the vagina and bladder.

Figure 3 panel 2. I would change the colors of the symbols. I can't tell the difference between genera that contain known uropathogenic species from those genera that are thought to consist mostly of non-pathogens; for example, the difference between *Pseudomonas* and *Lactobacillus*.

Reviewer #3 (Remarks to the Author):

In this revision, the authors have addressed several of the concerns previously raised by the reviewers. That said, the manuscript still lacks relevant methods-related details which would preclude others from repeating this work. I would encourage the authors to include these details with the level of specificity needed to support replication in the methods section and/or a supplement if limited by space.

Specific comments:

Pg 13, line 20 — Mean representation of each genus in the control was used to filter out potential contaminants. Please provide additional detail. Average read count?

Pg 15 — Antimicrobial resistance determination — this is still not detailed enough. It could not be replicated by another scientist and additional detail should be provided, at a minimum as a supplemental file.

Pg 15 — CARD db version should be specified.

Pg 5, line 31 — use of GRAMMy is not included in the methods section. The whole first paragraph of the "Infectome Screening" section should be in the methods section.

Pg 6, lines 22-32 Method used to calculate area under the curve of culture-based result vs. metagenomic results is not described, and the value of reporting AUC values in cases of small n (e.g., n=3 or less) is questionable.

Pg 7, line 20 — *Gardnerella* and *Ureaplasma* are commonly identified as a part of the vaginal microbiota but are generally not considered members of the healthy vaginal microbiota. Women who carry *Gardnerella* typically have bacterial vaginosis and are at risk for BVI and other infections.

Pg 8, Genome coverage — given the potentially pathogenic nature of *Gardnerella*, a *Lactobacillus* species (e.g., *iners*, *crispatus*, *johnsonii*) would be a better choice as an example of a typical

commensal of the vaginal tract.

Pg 9, Lines 20-33 — This type of information is what was asked for during the initial review and should be included in the methods section, along with additional detail regarding their implementation.

Point-by-point address of the specific comments raised by the reviewers.

(Original report text in italic; our report in blue font)

Reviewer #1 (Remarks to the Author):

The authors have been responsive to the reviewers and the result is a much better manuscript. I have only 3 suggestions.

We thank the reviewer for the careful reading of our manuscript and the many suggestions that have allowed us to significantly improve this paper.

1. *Page 4, line 33. "...fewer DNA fragments that are shorter" is ambiguous. Do you mean fewer shorter DNA fragments with no effect on longer fragments? I think not. It would be crystal clear if it read "...fewer DNA fragments that are [also] shorter. "*

We agree with the reviewer and have made this change in the text. All changes made to the manuscript are highlighted in yellow.

2. *Page 8. Line 31. "...typically commensal bacterial species, G. vaginalis...." I would be careful about calling G. vaginalis a commensal. In the vagina, it's a bit controversial; some studies find it associated with bacterial vaginosis, while others do not. In the adult female bladder (transurethral catheter-collected urines), it is associated with urgency urinary incontinence (Pearce et al., 2014). I would back off and call it a common inhabitant of the vagina and bladder.*

We agree with the reviewer and have changed the text referring to *G. vaginalis* as a common inhabitant, rather than a commensal.

3. *Figure 3 panel 2. I would change the colors of the symbols. I can't tell the difference between genera that contain known uropathogenic species from those genera that are thought to consist mostly of non-pathogens; for example, the difference between Pseudomonas and Lactobacillus.*

Another good suggestion, we have changed the colors of the symbols in this figure.

Reviewer #3 (Remarks to the Author):

In this revision, the authors have addressed several of the concerns previously raised by the reviewers. That said, the manuscript still lacks relevant methods-related details which would preclude others from repeating this work. I would encourage the authors to include these details with the level of specificity needed to support replication in the methods section and/or a supplement if limited by space.

We thank the reviewer for the careful reading of our manuscript. We followed the reviewer's suggestions and have included extensive additional methods-related detail in the new version of the manuscript. Changes made in the new version of the manuscript are highlighted in yellow. To support reproducibility further, we share primary data in the database for Genotypes and Phenotypes, and we provide all measurement data that was used to generate the figures in supplementary data tables.

Specific comments:

1. Pg 13, line 20 — Mean representation of each genus in the control was used to filter out potential contaminants. Please provide additional detail. Average read count?

We have improved the writing of this section, and provide additional detail.

2. Pg 15 — Antimicrobial resistance determination — this is still not detailed enough. It could not be replicated by another scientist and additional detail should be provided, at a minimum as a supplemental file.

We provide extensive detail on the procedures used at Weill Cornell Medicine for Antimicrobial Resistance Determination. See new version of the methods section.

3. Pg 15 — CARD db version should be specified.

We specify the CARD database version in the new version of the text.

4. Pg 5, line 31 — use of GRAMMy is not included in the methods section. The whole first paragraph of the “Infectome Screening” section should be in the methods section.

We have followed this suggestion and moved the beginning of this paragraph to the methods section.

5. Pg 6, lines 22-32 Method used to calculate area under the curve of culture-based result vs. metagenomic results is not described, and the value of reporting AUC values in cases of small n (e.g., $n=3$ or less) is questionable.

We provide details on the approach used to calculate AUC values in the methods section. We have removed AUC values for those cases where the number of culture positives available was very small (less than two). We note that the total number of samples used in all Receiver Operator Characteristic analyses was high (high number of culture negatives where available for all comparisons). Finally, in the new version of the text, we provide 95% Confidence Intervals for all the quoted AUCs.

6. Pg 7, line 20 — Gardnerella and Ureaplasma are commonly identified as a part of the vaginal microbiota but are generally not considered members of the healthy vaginal microbiota. Women who carry Gardnerella typically have bacterial vaginosis and are at risk for BVI and other infections.

We agree with the reviewer and have updated the text accordingly.

7. Pg 8, Genome coverage — given the potentially pathogenic nature of Gardnerella, a Lactobacillus species (e.g., iners, crispatus, johnsonii) would be a better choice as an example of a typical commensal of the vaginal tract.

Rather than exchange *G. vaginalis* for *Lactobacillus*, we have updated this section to denote that *G. vaginalis* can be pathogenic as the reviewer points out. See also comment 2, reviewer #1.

8. Pg 9, Lines 20-33 — This type of information is what was asked for during the initial review and should be included in the methods section, along with additional detail regarding their implementation.

In the methods section, we provide extensive detail on the methodologies used in the clinic for antimicrobial susceptibility testing. See also comment 2.